# Anti-Quorum-Sensing Potential of Ethanolic Extracts of Aromatic Plants from the Flora of Cyprus

**DOI:** 10.3390/plants11192632

**Published:** 2022-10-07

**Authors:** Tolis Panayi, Yiannis Sarigiannis, Elena Mourelatou, Evroula Hapeshis, Christos Papaneophytou

**Affiliations:** Department of Life and Health Science, School of Sciences and Engineering, University of Nicosia, 2417 Nicosia, Cyprus

**Keywords:** quorum sensing, autoinducer, organic plant extracts, biofilms, swimming motility, swarming motility

## Abstract

Quorum sensing (QS) is a form of intra- and inter-species communication system employed by bacteria to regulate their collective behavior in a cell population-dependent manner. QS has been implicated in the virulence of several pathogenic bacteria. This work aimed to investigate the anti-QS potential of ethanolic extracts of eight aromatic plants of Cyprus, namely, *Origanum vulgare* subsp. *hirtum*, *Rosmarinus officinalis*, *Salvia officinalis*, *Lavendula* spp., *Calendula officinalis*, *Melissa officinalis*, *Sideritis cypria*, and *Aloysia citriodora*. We initially assessed the effects of the extracts on autoinducer 2 (AI-2) signaling activity, using *Vibrio harveyi* BB170 as a reported strain. We subsequently assessed the effect of the ethanolic extracts on QS-related processes, including biofilm formation and the swarming and swimming motilities of *Escherichia coli* MG1655. Of the tested ethanolic extracts, those of *Origanum vulgare* subsp. *hirtum*, *Rosmarinus officinalis*, and *Salvia officinalis* were the most potent AI-2 signaling inhibitors, while the extracts from the other plants exhibited low to moderate inhibitory activity. These three ethanolic extracts also inhibited the biofilm formation (>60%) of *E. coli* MG1655, as well as its swimming and swarming motilities, in a concentration-dependent manner. These extracts may be considered true anti-QS inhibitors because they disrupt QS-related activities of *E. coli* MG1655 without affecting bacterial growth. The results suggest that plants from the unexplored flora of Cyprus could serve as a source for identifying novel anti-QS inhibitors to treat infectious diseases caused by pathogens that are resistant to antibiotics.

## 1. Introduction

Antibiotics have been used to prevent and treat bacterial infections in humans and animals; however, their inappropriate use has led to the development of multi-drug-resistant pathogens [1]. On the contrary, the identification of new antibiotics has steadily decreased since the 1970s [2], while many pharmaceutical companies have abandoned research on antibiotics [3]. Thus, the World Health Organization (WHO) has recently called antibiotic resistance “*an increasingly serious threat to global public health that requires action across all government sectors and society*” (http://www.who.int/mediacentre/factsheets/fs194/en, accessed on 10 June 2022). It is predicted that by 2050, 10 million deaths worldwide will be attributable to antimicrobial resistance [4]. Bacteria develop antibiotic resistance extremely quickly and share it with other bacteria [5]. Among bacteria’s different mechanisms for fighting antibiotics, the most threatening are those that use resistance genes on plasmids and integrons [6]. The information for resisting antibiotics is shared not only between individual bacteria of the same species, but also between species and, often, bacterial kingdoms [7]. Interestingly, Gram-negative (G^−^) bacteria can obtain antibiotic resistance genes from a shared pool [8].

Several bacteria regulate their behavior in a cell-density-dependent manner using a cell-to-cell signaling communication system called quorum sensing (QS) [9,10]. This mechanism regulates the expression of specific genes, and it is affected by bacterial cell population density. In the QS communication system, specific signaling molecules are released at low concentrations, and these molecules are effective only when the population of bacteria is high [11]. QS bacteria produce and secrete acyl-homoserine lactone (AHL)-signaling molecules, also known as autoinducers (AIs), which accumulate in the environment as the density of the bacteria cells increases [12]. When a threshold stimulatory concentration of AIs is reached, a signal transduction cascade is triggered, which eventually affects the behavior of the bacteria [13]. It has been demonstrated that QS plays pivotal roles in the regulation of virulence factors in several pathogens, as the release of AIs facilitates the transcription of specific genes involved in antibiotic resistance [14], biofilm formation [12,15], and swarming motility [16].

AHLs are produced by members of the LuxI family of AHL synthases and are mainly employed by G^−^ bacteria. On the contrary, Gram-positive (G^+^) bacteria lack LuxI or LuxR homologs; therefore, they use modified oligopeptides as AIs. In addition, a “universal” quorum sensing signal, the autoinducer-2 (AI-2), which is encoded by the *luxS* gene*,* has been identified in both G^−^ and G^+^ bacteria. Furthermore, the luxS-mediated QS (AI-2 signaling) is a universal communication system involved in the regulation of various behaviors in bacteria [17]. Notably, AI-2 is employed for interspecies communication between G^+^ and G^−^ bacteria and, thus, is of particular interest [18]. AI-2 is widely used as a target for screening potential anti-QS compounds by using the *Vibrio harveyi* bioassay, as previously described [19,20].

Considering the significance of QS during bacterial pathogenesis, research has focused on inhibiting QS. In contrast to antimicrobial compounds, anti-QS or ‘antipathogenic’ compounds do not cause cell death or growth inhibition [21]. Recent evidence revealed that specific anti-QS compounds could decrease the pathogenicity of bacteria and the formation of biofilms. Significantly specific anti-QS compounds increase the susceptibility of bacteria to antimicrobial drugs (e.g., antibiotics) and bacteriophages [22]. Several plant extracts and essential oils demonstrate anti-QS activity because their structures share similarities with the molecules employed in the QS communication system (e.g., AHL). Therefore, these natural anti-QS compounds can inhibit the AHL activity by competing with it [23]. Furthermore, several plant extracts disrupt the signal receptors (e.g., LuxR/LasR) of the AHL molecules [24]. Other natural extracts utilize a binary mechanism to interrupt QS signaling, i.e., they inhibit AHL activity and decrease the biosynthesis of AHLs by the bacteria [25]. The anti-QS potential of natural compounds has been reviewed elsewhere [26,27] and will not be discussed here.

Cyprus is located in the extreme north-eastern corner of the Mediterranean Sea, and consequently, the soil and climatic conditions might contribute to the large variety of plant chemotypes. The flora of Cyprus is rich in endemic taxa and comprises 1640 indigenous taxa (species and subspecies), 244 introduced taxa occurring in the wild, 42 hybrids, and 84 species with unclear status [16]. In addition, more than 650 medical plants have been identified in Cyprus. However, the antimicrobial properties of the medicinal and aromatic plants of Cyprus have not been extensively studied; therefore, they offer a unique collection of phytochemicals with novel microbial-disease-controlling potential. These plants and their extracts can open up the possibility of identifying novel quorum sensing inhibitors.

In this context, this work aimed to evaluate the anti-QS potential of eight plants’ ethanolic extracts, namely, *Origanum vulgare subsp. hirtum, Rosmarinus officinalis, Salvia officinalis, Lavendula spp., Calendula officinalis, Melissa officinalis, Sideritis cypria,* and *Aloysia citriodora* from the flora of Cyprus. We initially assessed the effects of the extracts on the AI-2 signaling activity of *E. coli* MG1655 using the well-established *V. harveyi* bioassay [19]. We further evaluated the effects of extracts against other bacterial functions related to QS, including biofilm formation and the swimming and swarming motilities. We also investigated the effects of ethanolic extracts on bacterial growth and viability. To the best of our knowledge, this is the first study to examine the anti-QS properties of extracts obtained from plants of the flora of Cyprus.

## 2. Results

### 2.1. Plant Material and Extraction of Active Compounds

Active compounds were extracted from eight plants of the flora of Cyprus using a sonication-assisted method (Table 1). We used ethanol as a solvent to extract the active compounds because it was the greenest and safest among the solvents found in the literature. In this work, we will use the common names of the plants (Table 1).

### 2.2. Inhibition of AI-2 Activity by the Ethanolic Extracts

We initially evaluated the anti-QS potential of the ethanolic extracts of the eight plants by monitoring the AI-2 inhibition using the well-studied *V. harveyi* bioassay [19]. In detail, AI-2 inhibition was determined by incubating the *V. harveyi* BB170 reporter strain with a known concentration of exogenous AI-2 (i.e., cell-free supernatant (CFS) from an *E. coli* culture) to induce luminescence and with either one of the ethanolic extracts or its respective blank medium (i.e., CFS without the extract). Inhibition was considered to have occurred when the luminescence of the sample was lower than that of its corresponding blank.

It was previously demonstrated that the AI-2 signaling molecule of *E. coli* reaches its maximum concentration at the mid-to-late growth phase, while a significant decrease in its concentration is observed at the stationary phase [28]. To investigate whether the release of the AI-2 signaling molecule was growth-dependent, we initially evaluated the levels of AI-2 production by *E. coli* MG1655 at various time points. *V. harveyi* BB120 was used as a positive control. The AI-2 activity was expressed as the fold activation compared to a sample that was not inoculated with the CFS medium (negative control). Figure 1 shows the fold induction of the AI-2 activity in CFSs collected from *E. coli* MG1655 grown in Luria Bertani (LB) medium supplemented with 0.5% glucose, as measured using the *V. harveyi* bioassay. The concentration of the AI-2 signaling molecule of *E. coli* MG1655 increased with the incubation time until 6 h; however, a decrease was observed after 6 h of growth. Based on these findings, for the subsequent experiments, we used the CFSs from *E. coli* cultures that were grown for 5 h at 37 °C in the presence of 0.5% glucose.

We subsequently tested the inhibitory effects of the ethanolic extracts from the eight plants (Table 1) at a final concentration of 2 mg/mL on the AI-2 signaling activity of *E. coli* MG1655. Due to the intense color of most of the ethanolic extracts (Table 1), concentrations higher than 2 mg/mL interfered with the downstream assays, especially with the motility assays and the evaluation of the bactericidal activity (as described in the following paragraphs). Using *V. harveyi* BB170 as the reported strain, the extracts from oregano and rosemary inhibited the AI-2 activity of *E. coli* MG1655 by 92.2 ± 1.6% and 93.5 ± 1.2%, respectively (Figure 2A). Moreover, the extracts from common sage inhibited the AI-2 activity by 67.1 ± 3.3%, whereas the extracts of the other five plants exhibited AI-2 inhibition ranging from approximately 7% to 45% (Figure 2A).

We further examined the effects of the three extracts that showed the highest AI-2 inhibition, i.e., oregano, rosemary, and common sage, by testing different concentrations ranging from 0.25 to 2.0 mg/mL. As illustrated in Figure 2B, all ethanolic extracts inhibited the AI-2 signaling activity of *E. coli* MG1655 in a concentration-dependent manner.

### 2.3. Effects of Plant Extracts on Biofilm Formation

It was previously demonstrated that QS plays a vital role in biofilm formation and differentiation [29,30]. Therefore, we subsequently tested the effects of ethanolic extracts of oregano and rosemary at a final concentration of 1.0 mg/mL and that of common sage at 2.0 mg/mL on the formation of *E. coli* MG16555 biofilms by using crystal violet staining. For comparison purposes, we selected the aforementioned concentrations of the three extracts because they produced a similar effect on AI-2 signaling activity (i.e., ~60% inhibition). The extracts of lavender, calendula, lemon balm, Cyprian siderites, and lemon beebrush were also tested at a final concentration of 2 mg/mL. As shown in Figure 3, the extracts from oregano, rosemary, and common sage exhibited the greatest inhibitory effects (>60%) on the biofilm formation of *E. coli* MG1655. Biofilm formation was less affected (26% and 19%, respectively) by the ethanolic extracts from lavender and lemon balm. In contrast, the extracts from calendula, Cyprian sideritis, and lemon beebrush had only a slight effect on biofilm formation by *E. coli* MG1655 (Figure 3).

### 2.4. Impacts of Phytochemicals on the Swarming and Swimming Motilities of E. coli MG 1655

The swarming and swimming motilities induced by QS are vital features of G^−^ bacteria for surface attachment during the early stages of the formation of biofilms and the subsequent maturation thereof [31]. Therefore, we investigated the effects of the eight ethanolic extracts on the motility of *E. coli* MG1655.

As shown in Figure 4, the ethanolic extracts of oregano, rosemary, and common sage significantly reduced both types of motilities. Importantly, these three ethanolic extracts inhibited both the swarming and swimming motilities in a concentration-dependent manner (Appendix A). The ethanolic extracts of the other five plants had either slight or no effects on both types of motilities of *E. coli* MG1655 (Figure 4).

In detail, at the highest concentrations tested, the ethanolic extracts of oregano and rosemary (both at 1 mg/mL) as well as that of common sage (2 mg/mL), significantly (*p* < 0.0001) inhibited the swarming motility of *E. coli* MG1655 by 54.7%, 58.3%, and 48.4%, respectively (Figure 4A1,A2). The ethanolic extracts of lavender, calendula, and lemon balm had only a limited effect on swarming motility (~10% inhibition), but these differences did not reach statistical significance. On the other hand, we could not evaluate the effect of the ethanolic extracts of Cyprian sideritis and lemon beebrush on the swarming motility of *E. coli* MG1655 because diffused zones were observed (Figure 4A1). Regarding the swimming motility, the highest inhibition (45.7%; *p* < 0.001) was recorded in the presence of the ethanolic extract of rosemary at 1 mg/mL. The ethanolic extracts of oregano (1 mg/mL) and common sage (2 mg/mL) inhibited the swimming motility of *E. coli* MG1655 by 42.3% (*p* < 0.0001) and 17.2% (*p* < 0.0001), respectively (Figure 4B1,B2). However, we could not evaluate the effects of the ethanolic extracts of the other five plants on the swimming motility of *E. coli* MG1655 because huge diffused zones were formed (Figure 4B1). Inhibition of both types of motilities by the ethanolic extracts of oregano, rosemary, and common sage can be associated with the reduced ability of *E. coli* to form biofilms in the presence of the aforementioned ethanolic extracts.

### 2.5. Effects of Plant Extracts on the Growth of E. coli

To verify that none of the observed anti-QS activities of the ethanolic extracts of oregano, rosemary, and common sage were correlated with bactericidal activity, the effects of the extracts on the growth of *E. coli* MG1655 were examined. Interestingly, none of the three ethanolic extracts exhibited any bactericidal activity, as determined by the inhibition-of-growth assay and viable plate counts (Appendix A). Therefore, the ethanolic extracts from oregano, rosemary, and common sage can be considered “true” QS inhibitors that do not rely upon the antibacterial activity of traditional antibiotics [32].

### 2.6. Determination of the Composition of Ethanolic Extracts with the Highest Anti-Quorum-Sensing Activity by LC-MS

LC-MS analysis revealed five major active components (Table 2 and Figure 5) in the three ethanolic extracts with the highest anti-QS activity. In detail, our analysis revealed that the extracts of oregano, rosemary, and common sage shared three common active compounds: carnosol, chlorogenic acid, and quercetin. Interestingly, in the ethanolic extract of oregano, we also detected apigenin and rosmarinic acid among the major active components; however, we did detect these compounds in the extracts obtained from rosemary or common sage (Table 2 and Appendix A). Further experiments for determining other active compounds in the three ethanolic extracts are currently in progress in our laboratory.

## 3. Discussion

Antibiotics are essential in preventing and treating bacterial infections [33]. Unfortunately, under selective pressure from antibiotics, bacteria have developed sophisticated mechanisms to fight these drugs, leading to the development of strains that are resistant to antibiotics [34]. The development of antibiotic resistance by microorganisms, including bacteria, is a major global health issue (reviewed in [35]). To this end, the scientific community’s attention has been turned toward the identification of antipathogenic drugs that do not kill bacteria, and thus, bacteria do not undergo the selective pressure that leads to the development of strains that are resistant to antibiotics [36]. Therefore, it could be possible to inhibit the virulence of pathogenic bacteria without killing cells, and such antipathogenic compounds may be used alone or in combination with antibiotics [37].

It has been demonstrated that both G^−^ and G^+^ bacteria employ a QS communication system to regulate gene expression in a cell-density-dependent manner [38]. When bacteria reach a critical concentration, they release signal molecules called AIs [37]. QS is often employed to regulate beneficial genes expressed by a bacterial community, including genes implicated in virulence, biofilm formation, swarming and swimming motilities, stress resistance, and antibiotic resistance [39,40]. Therefore, inhibition of QS communication among bacteria could be used as an alternative strategy to fight multi-drug-resistant bacteria, while any compound that is able to inhibit AI activity without interfering with the growth rate of bacteria can be considered to be a potential QS inhibitor [37]. It has been reported that several plant extracts and essential oils exhibit antimicrobial and anti-QS activity; therefore, identifying anti-QS compounds from natural sources, including aromatic plants, is of particular interest in the scientific community [41]. The essential oils of several plants have demonstrated promising anti-biofilm-formation and anti-QS activities [42]. As mentioned before, various mechanisms could achieve the inhibition of the QS communication system by using natural compounds [25,27,43], including (i) inhibition of the biosynthesis of QS signaling molecules, (ii) competitive inhibition, i.e., some natural molecules share structural similarities with the QS signaling molecules and, thus, compete for binding to corresponding receptor proteins affecting the signal transduction pathway in QS, and (iii) inhibition of QS signal reception by acyl homoserine lactone. Nevertheless, further studies are required to elucidate the mechanisms of the anti-QS activities of natural products.

In this work, we examined the anti-QS activity of the ethanolic extracts of eight aromatic plants from the flora of Cyprus (Table 1). The anti-QS activity of the ethanolic extracts was assessed using various bioassays, including inhibition of AI-2 signaling activity and biofilm formation, as well as motility assays. Screening of ethanolic extracts for inhibition of AI-2 activity was performed using the widely used *V. harveyi* assay, with BB170 as a reported strain. *V. harveyi* BB170 is exquisitely sensitive to AI-2 (it has the QS phenotype AI-1^−^, AI-2^+^); therefore, even low amounts of AI-2 can be detected using this bioassay. Inhibition was considered when the luminescence of a tested compound (i.e., ethanolic extract) was lower than that of the respective blank control. Our preliminary results revealed *E. coli* MG1655 exhibited significant AI-2 activity in an LB supplement with 0.5% glucose after 5 h cultivation, while the signaling activity was comparable to that of *V. harveyi BB120* (AI-1^+^/AI-2*^+^)* (Figure 1). It should be pointed out that when *E. coli* is grown in LB containing glucose, the sugar inhibits the uptake of AI-2 into the cells and, thus, accumulates in the culture supernatant [43,44]. Subsequently, our preliminary screening revealed that the ethanolic extracts of oregano and rosemary at 2 mg/mL exhibited the highest inhibition of AI-2 activity (>90%) (Figure 2A). Notably, the ethanolic extracts of the three plants had a concentration-dependent effect on AI-2 signaling activity (Figure 2B).

Previous studies have highlighted the anti-QS potential of extracts, essential oils, and other single bioactive compounds of oregano [26,45,46,47] and rosemary [48,49]. The composition of bioactive molecules of the extracts depends on the extraction technique used, but also on the part of the plant used [50,51,52]. Furthermore, a plant’s intraspecies variations in bioactive compound composition could be due to several factors, including the extraction and analytical method used, soil and environmental conditions, and other plant factors [53]. Full identification, characterization, and quantification of the components of the extracts were out of the scope of this article. However, our LC-MS analysis of the extracts obtained from oregano, rosemary, and common sage identified that there are multiple phenolic components, which can be divided into two main groups: phenolic acids (rosmarinic acid and chlorogenic acid) and flavonoids (apigenin, quercetin, or luteolin). In addition, diterpenes, such as carnosol, were also detected. Carnosol, chlorogenic acid, and querqetin were detected in all of the plant extracts, whereas rosmarinic acid and apigenin were detected only in oregano (Table 2). Similar results for oregano have been reported by Exarchou and co-workers [54], while rosmarinic acid, quercetin, and apigenin were detected in ethanolic extracts of oregano from Taiwan [55]. It has been demonstrated that carnosic acid and carnosol inhibit the QS-related process of *Staphylococcus aureus*, including its virulence [49]. The anti-QS potential of rosmarinic acid [56], quercetin [57], and chlorogenic acid [58] have also been reported. To the best of our knowledge, the anti-QS potential of apigenin has not been elucidated. In rosemary extracts, high concentrations of carnosol were identified. High levels of carnosol and carnosic acid have also been reported in rosemary extracts from Tunisia [52] and Morocco [59], while common sage (*Salvia officinalis)* extracts from Finland [60] exhibited low levels of carnosic acid and carnosol. Carnosol and carnosic acid are degraded rapidly with long extraction times and at high temperatures. It would be useful for further exploration and exploitation to collect and compare the bioactive profiles of these plants located in the Mediterranean Basin under the same conditions. Furthermore, despite the antimicrobial properties of extracts of common sage (*Salvia officinalis*) that have been previously reported [51], their anti-QS potential remains inconclusive. Our initial screening revealed that the ethanolic extract of common sage inhibited AI-2 activity in a concentration-dependent manner (Figure 2B). In contrast, at the highest concentration tested (2 mg/mL), a 65% inhibition of AI-2 activity was recorded (Figure 2A,B). Unfortunately, due to the intense color of the ethanolic extract of common sage (Table 1), concentrations higher than 2 mg/mL interfered with the motility assays and growth inhibitions assays, whereas the precipitation of the extracts was observed at concentrations ≥5 mg/mL in aqueous solutions, which was probably due to their hydrophobic nature. Nevertheless, the similar chromatographic profiles for common sage and rosemary extracts (Table 2) may explain their similar effects on the anti-QS-related process of *E. coli* MG1655. The ethanolic extracts of lavender, calendula, lemon balm, Cyprian sideritis, and lemon beebrush inhibited the AI-2 activity by less than 45% (Figure 2A).

We evaluated the anti-QS activities of the eight extracts (Table 1) using concentrations that produced comparable AI-2 inhibition (~60%), i.e., 1 mg/mL for oregano and rosemary and 2 mg/mL for common sage. The ethanolic extracts of the other five plants were also tested at 2 mg/mL. We subsequently tested the effects of the eight ethanolic extracts on the formation of biofilms by *E. coli* MG1655 by using crystal violet staining. QS has been implicated in the development of biofilms in both G^−^ and G^+^ species, while biofilm formation is one of the strategies employed by bacteria for developing resistance to antibiotics [61]. In addition, treating diseases caused by bacteria that form biofilms requires prolonged treatment, which may lead to antibiotic resistance due to high evolutionary pressure [62]. Herein, the extracts of oregano, rosemary, and common sage significantly inhibited the formation of biofilms by *E. coli* MG1655 (Figure 3) without affecting the bacterial growth (Appendix A). The ability of extracts and essential oils of oregano to inhibit the formation of biofilms by *Candida spp.* [63]*, Staphylococci,* and *E. coli* [64] has been previously reported. Likewise, the inhibitory effect of rosemary extracts on biofilm formation by various pathogenic bacteria, including *Candida albicans*, *Staphylococcus aureus*, and *Pseudomonas aeruginosa,* has been described [65]. Recently, Selim et al. [66] reported the antibiofilm potency of the essential oil of common sage (*Salvia officinalis* L) against antibiotic-resistant *Salmonella enterica.* The potential of extracts of common sage to inhibit biofilm formation by *P. aeruginosa* has also been reported [67].

QS-dependent swimming motility, which is driven by flagella, is vital for the initiation of cell/surface attachment during biofilm formation [68]. In this work, we demonstrated that the extracts of oregano, rosemary, and common sage inhibited the swimming motility of *E. coli* MG1655 in a dose-dependent manner (Figure 4 and Appendix A). In addition to the swimming migration, swarming motility, another QS-dependent motility, has been implicated in biofilm formation [69]. Our results revealed a dose-dependent inhibition of the swarming motility of *E. coli* MG1655 by the extracts obtained from the three plants mentioned above (Figure 4 and Appendix A). The correlation of inhibition of biofilm formation with the reduced swimming and swarming motilities of a variety of bacterial pathogens in the presence of different extracts of plants and fruits, including *Capparis spinosa* [70] and *Salvadora persica* L. [71], as well as in the presence of clove oil, has been reported [72]. To the best of our knowledge, this is the first study to examine the effects of extracts of oregano, rosemary, and common sage on both types of motilities of *E. coli* MG1655.

## 4. Materials and Methods

### 4.1. Selection and Preparation of Plants

A total of eight plants (Table 1) were collected from the Cypriot National Agricultural Department, Nicosia, Cyprus. Selection and collection of plants were carried out based on good plant authentication and identification practices (GPAIPs) and good agricultural and collection practice (GACPs) [73]. The collected plants were handled with standard storage protocols and transported by being wrapped in plastic bags. The plants were washed thoroughly under running tap water, rinsed with ddH_2_O, air-dried at 25 °C under shade, cut to the appropriate size, packed in plastic bags, and kept until extraction.

### 4.2. Extraction of Active Compounds from Plants

For the extraction of active compounds from the eight plants, we employed alternative methods that minimized the use of solvents and enabled process intensification for the cost-effective production of high-quality extracts. The REACH (Registration, Evaluation, Authorization, and Restriction) directive limited the use of several chemical solvents and reagents in extraction or industrial manufacturing products (https://echa.europa.eu/regulations/reach/understanding-reach, accessed on 12 May 2022). In this work, we followed the “Six Principles of Green Extraction of Natural Products” [74], which are: “1: Innovation by the selection of varieties and use of renewable plant resources; 2: Use of alternative solvents and/or water or agro-solvents; 3: Reduce energy consumption by energy recovery and using innovative technologies; 4: Production of co-products instead of waste; 5: Reduce unit operations and favor safe, robust and controlled processes; 6: Aim for a non-denatured and biodegradable extract without contaminants”. 

Based on the abovementioned principles, we used ethanol as a solvent to extract the active compounds. Instead of using the traditional energy- and time-consuming Soxhlet methodology, we used a sonication-assisted technology, which saved time and reduced energy consumption. Our preliminary experiments revealed that the best ratio (dry plant/volume) to obtain the maximum extract per dry mass was 1 g per 20 mL of solvent. Lower quantities, i.e., 0.5 g/10 mL of solvent, did not result in similar amounts, demonstrating a cut-off in the method. As a result, 1 g of dry material (plant) was mixed with 20 mL of ethanol in a 50 mL centrifuge vial. The mixture was sonicated in an ultrasonic water bath (Grant, UK) for 45 min at 45 °C and 200 W at 32–38 KHz. Subsequently, the solution was filtered through a 0.25 μm filter, and the solvent was removed under vacuum at 45 °C in rotavapor Buchi R-210. The extraction yields are summarized in Table 1. The residual extracts were resuspended in DMSO that was previously filtered with a 0.2 μΜ syringe filter (VWR, West Chester, PA, USA).

### 4.3. Bacterial Strains, Media, and Culture Conditions

*Escherichia coli* MG 1655 *(*ATCC-700926), *Vibrio harveyi* BB-120 (ATCC-BAA-1116), and *V. harveyi* BB-170 (ATCC-BAA-1117) were obtained from the American Type Culture Collection (ATCC; Wesel, Germany).

*E. coli* was grown in Luria Bertani (LB) medium consisting of 1% tryptone, 0.5% yeast extract, and 1% NaCl at 37 °C. *V. harveyi* BB-120 and BB-170 were grown at 30 °C in autoinducer bioassay (AB) medium (ATTC medium: 2034) consisting of (per 1 L) 17.53 g of ΝaCl, 6.02 g of MgCl_2_, and 2.0 g of casamino acids (vitamin-free). The pH of the medium was adjusted to pH 7.0 with 1 M KOH and autoclaved at 121 °C for 15 min. The solution was cooled to room temperature, and 10 mL of 1 M potassium phosphate buffer, pH 7.0, 10 mL of 0.1 M sterile arginine solution, and 20 mL of 50% sterile glycerol were added to the medium.

### 4.4. Autoinducer-2 Bioassay

The AI-2 bioassay was carried out as previously described [75,76]. The assay was based on the ability of the reported stain *V. harveyi* BB170 to specifically bioluminesce in response to AI-2. At lower cell densities of BB170 (10^6^–10^7^ CFU/mL), the bioluminescence could be recorded in response to the added (spiked) AI-2 [77].

#### 4.4.1. Preparation of Cell-Free Supernatants

*E. coli* was grown overnight at 37 °C in LB medium supplemented with medium containing 0.5% glucose. The next day, the overnight culture was used to inoculate (1:100) fresh LB medium containing 0.5% glucose, and the cultures were incubated at 37 °C for various times, as indicated in the text, under continuous shaking at 250 rpm. Cell-free supernatants (CFSs) were prepared by centrifuging the culture at 16,000 g for 15 min at 4 °C. The resulting supernatants were passed through 0.2 μm syringe filters (VWR, West Chester, PA), aliquoted, and stored at −20 °C until the AI-2 bioluminescence assay was carried out. CFSs containing *V*. *harveyi* AI-2 were prepared from *V*. *harveyi BB120* (AI-1^+^, AI-2^+^) and used as positive controls. In brief, *V. harveyi* BB120 was grown overnight at 30 °C in AB medium under continuous shaking. The CFSs were recovered from the overnight culture as described above for *E. coli*.

#### 4.4.2. Inhibition of AI-2 by the Ethanolic Extracts

The reported *V*. *harveyi* BB170 strain was grown for 16 h at 30 °C in AB medium and subsequently diluted (1:5000) into fresh AB medium. A total of 90 μL of the diluted cells were added into the wells of a 96-well plate and mixed with 10 μL of *E. coli* MG1655 or *V. harveyi* BB120 (for the screening experiments) or 9 μL of CFSs of *E. coli* MG1655 and 1 μL of each of the ethanolic extracts (of various concentrations, as described in the results). In addition, blank controls (9 μL of CFSs + 1 μL of DMSO) and a negative control (9 μL of AB medium + 1 μL of DMSO) were included in each experiment.

The plates were incubated at 30 °C under continuous shaking (100 rpm), and luminescence readings (in relative light units/RLU) were recorded every 20 min using a Perkin Elmer VictorX3 2030 Multiplate reader (PerkinElmer, Waltham, MA) in the chemiluminescence mode. The inhibition of AI-2 activity was expressed as a percentage relative to the blank control and was calculated using the following equation (Equation (1)) [78]:(1)$$\%\mathrm{AI}2\;\mathrm{inhibition}=\left(1-\frac{\mathrm{RLU}\;\mathrm{of}\;\mathrm{sample}}{\mathrm{RLU}\;\mathrm{of}\;\mathrm{blank}\;\mathrm{control}}\right)\text{×}100\;$$

### 4.5. Inhibition of Biofilm Formation

The effects of the plant extracts on biofilm formation were assessed in sterile 96-well flat-bottom polystyrene plates as previously described [79], with some modifications. Positive controls (bacteria cells + LB), medium controls (LB only), and solvent controls (cells + LB + DMSO) were used. All experiments were carried out in triplicate.

We added the appropriate concentration of plant extract to the test wells before inoculation. Plates were incubated at 37 °C under continuous shaking (100 rpm). After 48 h of cultivation, the content of each well was discarded, rinsed three times with phosphate-buffered saline (PBS), and fixed by drying for 1 h at 37 °C in the incubator. When the wells were fully dry, 200 μL of 0.1% crystal violet stain were added to each well and incubated for 15 min at 25 °C. The excess dye was rinsed off using tap water, and subsequently, 200 μL of 96% ethanol was added to the wells. The stain adhering to the biofilm biomass was recovered with ethanol, transferred to clean wells of 96-well plates, and the absorbance at 570 nm (A570 nm) was measured in a Perkin Elmer VictorX3 2030 Multiplate reader (PerkinElmer, Waltham, MA). The biofilm inhibition rate was defined using the following equation (Equation (2)):(2)$$\%\mathrm{Biofilm}\;\mathrm{inhibition}=\left(1-\frac{A570\;\mathrm{nm}\;\mathrm{of}\;\mathrm{the}\;\mathrm{sample}}{A570\;\mathrm{nm}\;\mathrm{of}\;\mathrm{the}\;\mathrm{positive}\;\mathrm{control}}\right)\text{×}100\;$$

All crystal violet assays were run in triplicate, with a minimum of three replicates per assay.

### 4.6. Motility Assays

Swimming and swarming motility assays were carried out as previously described [70,80], with some modifications. Both the swimming and swarming motility assays were conducted in the wells of 6-well plates (5 mL per well). Overnight cultures (2 µL; ~10^7^ CFU/mL) of *E. coli* in LB medium were point inoculated in swarming agar consisting of 1% (*w*/*v*) tryptone, 0.5% (*w*/*v*) yeast extract, 0.5% (*w*/*v*) NaCl, and 0.5% (*w*/*v*) agar with different concentrations of the ethanolic extracts, as described in the “*Results*” section. To assess the effects of ethanolic extracts on the swimming motility of *E. coli*, 2 μL (~10^7^ CFU/mL) from an overnight culture of the bacteria was point inoculated at the center of the wells of the 6-well plates containing 1% (*w*/*v*) tryptone, 0.5% (*w*/*v*) NaCl, and 0.3% (*w*/*v*) agar, as well as one of the ethanolic extracts at a final concentration of 1.0 or 2.0 mg/mL. Swimming and swarming motility wells containing none of the extracts were used as controls. Plates were incubated at 37 °C in the upright position for 16 h. The swimming and swarming migrations were recorded by measuring the diameters of the swim zones or swarm fronts, respectively, of the bacterial cells after incubation.

### 4.7. Effects of Ethanolic Extracts on Bacterial Growth

The effects of the ethanolic extracts on the growth of *E. coli* MG1655 were evaluated in liquid culture (200 μL) in the wells of a 96-well plate. Serial dilutions were performed to examine the effects of the ethanolic extracts of oregano, rosemary, and common sage at final concentrations ranging from 0.5 to 2 mg/mL after 20 h of cultivation. Growth controls (bacteria cells + LB), medium controls (LB only), and solvent controls (cells + LB + DMSO) were used. All experiments were carried out in triplicate. Optical density values at 600 nm (OD_600 nm_) were obtained using a Perkin Elmer VictorX3 2030 Multiplate reader (PerkinElmer, Waltham, MA, USA) at 0 and 20 h post-inoculation. To account for the effect of the extract color (bright green to very dark green) on the OD_600 nm_, the following formula (Equation (3)) was used [81]:(3)$$\%\;\mathrm{inhibition}=(1\;-\left(\frac{{\mathrm{OD}}_{t20h}-{\;\mathrm{OD}}_{t0h}}{{\mathrm{OD}}_{\mathrm{gc}20h}-{\;\mathrm{OD}}_{\mathrm{gc}0h}}\right))\;\text{×}\;100$$
where OD_t__20h_ is the optical density (600 nm) of the test well at 20 h post-inoculation, OD_t__0h_ is the optical density (600 nm) of the test well at 0 h post-inoculation, OD_gc__20h_ is the optical density (600 nm) of the growth control well at 20 h post-inoculation, and OD_gc__0h_ is the optical density (600 nm) of the growth control well at 0 h post-inoculation.

The effects of the ethanolic extracts of oregano, rosemary, and common sage on bacterial growth were further assessed by performing viable plate counts as previously described [80], with some modifications. Ethanolic extracts at a final concentration of 1 or 2 mg/mL and cultures of *E. coli* (10^7^ CFU/mL) were added to the wells of a 96-well plate (200 µL per well). Bacteria (10^7^ CFU/mL) without any of the extracts were used as a control. The plates were incubated at 37 °C without shaking for 24 h, and subsequently, bacterial suspensions were transferred to clean Eppendorf tubes, centrifuged at 5000 g for 5 min at 4 °C, washed three times with PBS, and resuspended in 200 μL of fresh LB medium. Each suspension was subsequently serially diluted in LB and plated on LB agar. After incubation at 37 °C for 16 h, the number of viable bacteria was determined and expressed as CFU/mL.

### 4.8. Determination of the Composition of Ethanolic Extracts with the Highest Anti-Quorum-Sensing Activity by HPLC and ESI-MS

The dried extracts were dissolved in 2 mL of DMSO, filtrated through, and analyzed with high-performance liquid chromatography (Alliance HPLC e2695, PDA 2998, Waters, Milford, MA, USA) and electrospray ionization mass spectrometry (ACQUITY QDa Mass Detector, Waters, Milford, MA, USA). The chromatographic separation was achieved with a column (Symmetry C18, 150 × 4.6 mm, 5 μm) at a flow of 0.8 mL/min with a linear gradient system for 30 min (A: 0.1% formic acid in Water, B: methanol). The column temperature was adjusted to 40 °C, and the injection volume was 5 μL. The exact analytical conditions are described in Appendix A. For mass detection, the positive-ion ESI mode was used. The rest of the capillary voltage settings were: Pos: 0.8 kV, gain: 1, and probe: 600 °C, while the cone voltage was set to 15 V.

### 4.9. Statistical Analysis

Unless otherwise stated, experiments were carried out in triplicate, and the data are presented as the mean values ± standard deviation (SD). One-way ANOVA followed by Tukey’s multiple-comparison test was used to compare the effects of (i) the extracts’ concentration on AI-2 activity (Figure 2) and (ii) the extracts on the swarming and swimming mobilities (Figure 4) of *E. coli* MG1655. Statistical significance was set at *p* < 0.05. The statistical analysis was performed using GraphPad Prism (v.8.2, GraphPad Software Inc., San Diego, CA, USA).

## 5. Conclusions

The trend of using natural compounds as QS inhibitors is gradually becoming an attractive approach in the field of developing new drugs to fight antibiotic-resistant bacteria. In this work, we identified three ethanolic extracts from endemic plants of Cyprus that significantly inhibited AI-2 signaling activity. The AI-2 molecule is of particular interest because it is a universal interspecies signaling molecule. Thus, inhibition of AI-2 could be a potential strategy for controlling bacterial pathogenicity. Biofilm formation, which QS also regulates, is one of the biggest challenges for human/animal health and the food industry. The three extracts also inhibited the formation of biofilm in *E. coli* MG1655 and the swimming and swarming motilities of the bacteria. Several natural products, including organic extracts of aromatic plants, display promising anti-QS activities by preventing biofilm formation and bacterial motility; thus, they could reduce the virulence and pathogenicity of antibiotic-resistant bacteria. The extracts identified in this work could be a starting point for further optimization and identification of novel anti-QS agents for the treatment of biofilms.

To conclude, this work identified a pool of potential anti-QS inhibitors that do not affect bacterial growth. Notably, antipathogenic compounds were identified, i.e., molecules that reduced the virulence of bacteria without killing them, and they did not impose selective pressure on the development of resistant strains. Further experiments and analyses of extracts’ compositions in bioactive compounds are required to elucidate the mechanism(s) by which they inhibit the QS activity in bacteria. Identifying chemical compounds (or specific compositions) with an anti-QS activity that are unique to the plants of the area of Cyprus would be particularly interesting.

## Figures and Tables

**Figure 1 plants-11-02632-f001:**
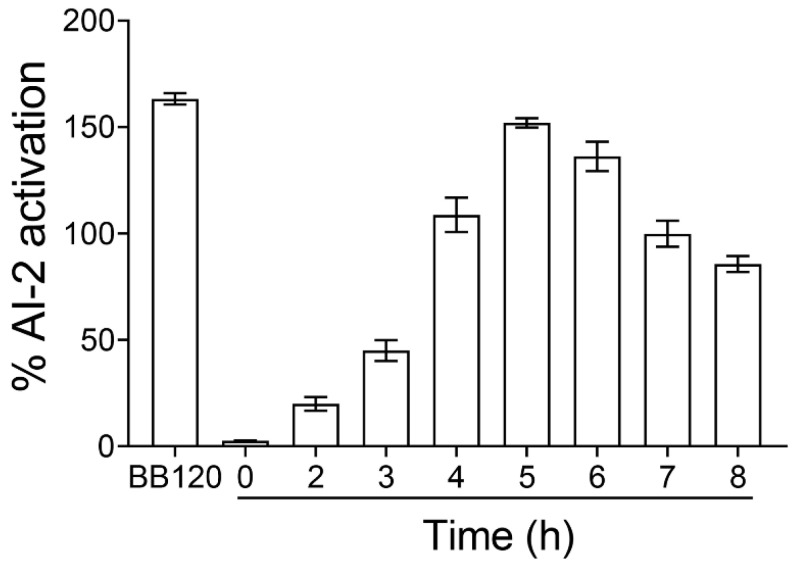
Time course of the AI-2 signaling activity in *E. coli* MG1655. *E. coli* was grown in LB medium supplemented with 0.5% glucose at 37 °C. At the indicated times, cell-free supernatants (CFSs) were prepared and assayed for AI-2 activity. The AI-2 signaling activity is presented as the percent activation compared to the non-inoculated negative control with CFS. A CFS obtained from an overnight culture of *V. harveyi* BB120 (AI-1^+^, AI-2^+^) was used as a positive control.

**Figure 2 plants-11-02632-f002:**
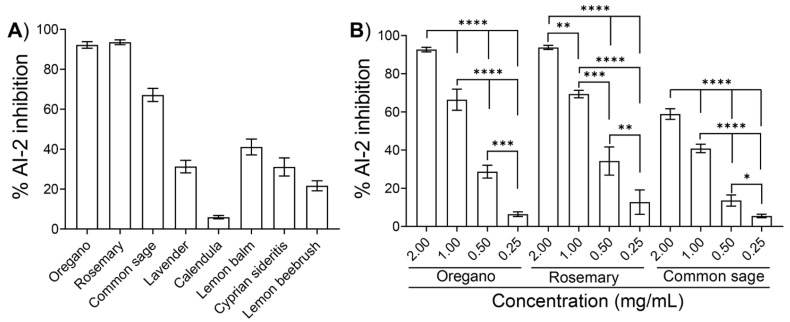
Effects of ethanolic extracts on the AI-2 activity of *E. coli* MG1655**.** Cell-free supernatants were collected from an *E*. *coli* culture in LB medium supplemented with 0.5% glucose after 5 h of cultivation at 37 °C and assayed for AI-2 activity in the presence of (**A**) one of the indicated plant extracts at a final concentration of 2 mg/mL or (**B**) different concentration of extracts obtained from oregano, rosemary, or common sage. The AI-2 signaling activity is presented as the percent inhibition compared to that of samples containing none of the extracts (blank control). The values are the means of the results of three independent experiments. Error bars indicate standard deviations. In (**B**), an ANOVA followed by Tukey’s multiple-comparison test was used for statistical analysis. Statistically significant differences are indicated with asterisks: * *p* < 0.05, ** *p* < 0.01, *** < 0.001 **** *p* < 0.0001.

**Figure 3 plants-11-02632-f003:**
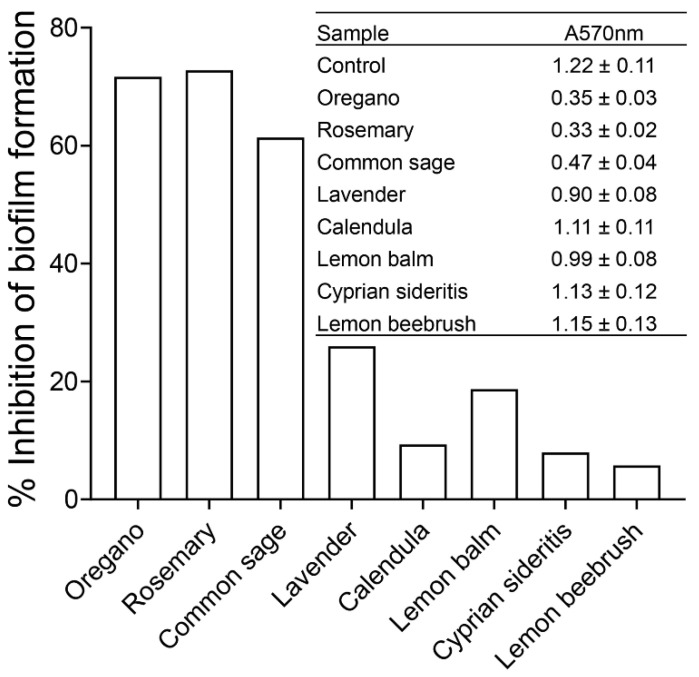
Effects of the ethanolic extracts from the eight aromatic plants from the flora of Cyprus on biofilm formation by *E. coli* MG1655 as quantified by crystal violet staining and measuring at A570 nm. Data are presented as the percentage of inhibition of biofilm formation compared to the control containing none of the ethanolic extracts**.** Inset: Absorbance values at 570 nm ± standard deviation following the crystal violet staining in three independent experiments.

**Figure 4 plants-11-02632-f004:**
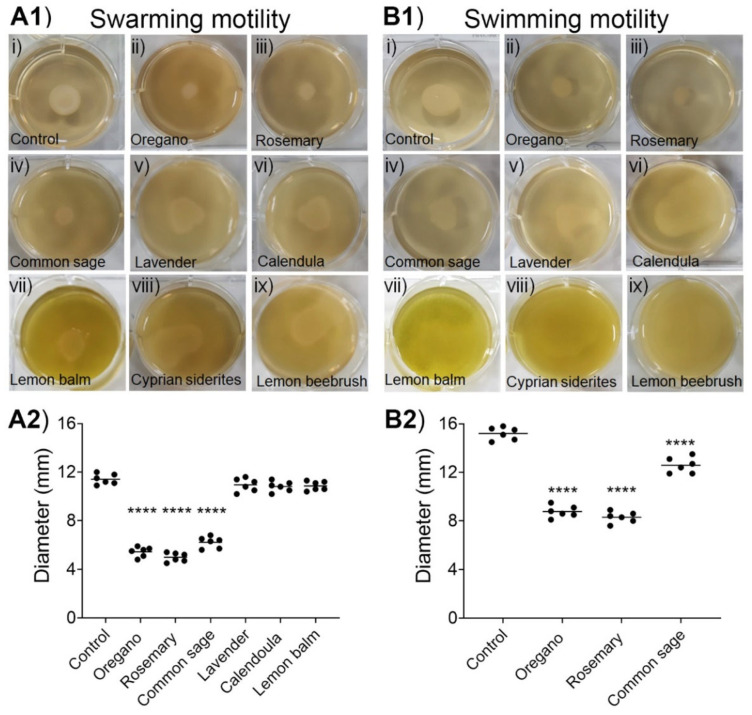
Inhibition of the swarming (**A1**) and (**A2**) and swimming (**B1**) and (**B2**) motilities of *E. coli* MG1655 by the ethanolic extracts**.** In panels (ii, iii) and panels (iv–ix), *E. coli* MG1655 was grown on LB agar containing 1 or 2 mg/mL, respectively, of the indicated ethanolic extract. In (**A2**) and (**B2**), horizontal bars indicate the mean values of six independent experiments. ANOVA followed by Tukey’s multiple-comparison test was used for statistical analysis. Only statistically significant differences are shown, and they are indicated with asterisks: **** *p* < 0.0001.

**Figure 5 plants-11-02632-f005:**
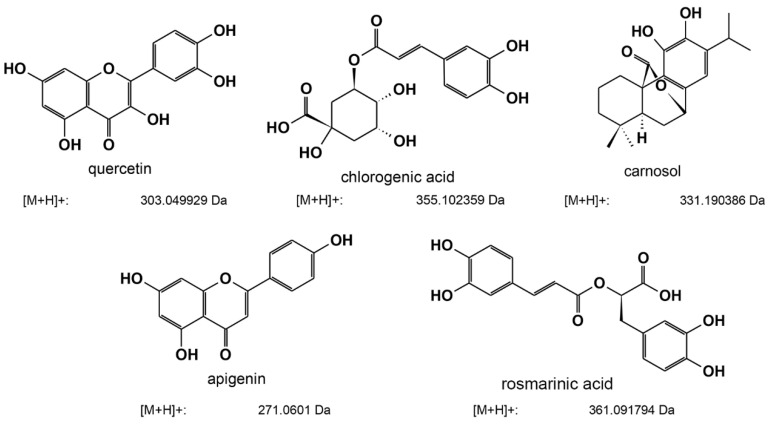
Chemical structures of the active compounds detected in the ethanolic extracts of oregano, rosemary, and common sage through LC-MS.

**Table 1 plants-11-02632-t001:** Plants used in this study and the recovery yields of their ethanolic extracts.

Scientific Name	Common Name	Family	Extraction Yield (% Dry Mass)	Extract Color Intensity ^1^
*Origanum vulgare subsp. hirtum*	Oregano	*Lamiaceae*	13.92	+
*Rosmarinus officinalis*	Rosemary	*Lamiaceae*	30.61	+
*Salvia officinalis*	Common sage	*Lamiaceae*	23.37	+++
*Lavendula spp*	Lavender	*Lamiaceae*	9.25	++
*Calendula officinalis*	Calendula	*Asteraceae*	18.32	++
*Melissa officinalis*	Lemon balm	*Lamiaceae*	12.98	++++
*Sideritis cypria*	Cyprian sideritis	*Lamiaceae*	14.87	++++
*Aloysia citriodora*	Lemon beebrush	*Verbenaceae*	7.20	+++

^1^ Color intensity of the ethanolic extracts in DMSO: +: bright green; ++: green; +++ dark green; ++++: very dark green.

**Table 2 plants-11-02632-t002:** Composition of ethanolic extracts of oregano, rosemary, and common sage as assessed by HPLC and ESI-MS.

Compound	HPLC Retention Time (min)	ESI-MS [M + H] ^+^_,_ Da	Plant
Quercetin	16.3	303.044929	Oregano Rosemary Common sage
Chlorogenic Acid	20.3	335.1022359	Oregano Rosemary Common sage
Carnosol	25.6	331.190386	Oregano Rosemary Common sage
Apigenin	15.2	271.0601	Oregano
Rosmarinic acid	16.7	361.091794	Oregano

## Data Availability

Not applicable.

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
