# Peer review of "Anti-Quorum-Sensing Potential of Ethanolic Extracts of Aromatic Plants from the Flora of Cyprus"

_plants, 2022, doi:10.3390/plants11192632_

Round 1
Reviewer 1 Report
This paper reports what is essentially a preliminary study of the effect of certain common Cyprus plant ethanol extracts on bacterial quorum sensing. Three of the extracts significantly inhibited AI-2 signaling activity, the formation of biofilm, and the swimming and swarming motility of the bacterium. These results are worthy of publication but it is important to follow up this study with the identification of the chemical components responsible for the observed activity. Some minor editorial work needs to be done on the text.
Reviewer 2 Report
The paper by Tolis Panayi et al. describes the anti-QS activity of some medicinal plants that originated from Cyprus against two model microorganisms, Vibrio harveyi BB170 and Escherichia coli MG1655.
The authors have performed classical experiments to show the anti-QS activities, with the highest exhibited by extracts from rosemary, common sage, and oregano. The authors have performed multiple experiments to assess QS-related processes, such as V. harveyi bioassay, assessment of residual biofilm by CV staining, and motility assays. Unfortunately, the work does not significantly contribute to the field; all the plants studied here have already been shown to inhibit QS-related processes. There is no evidence that rosemary, common sage, and oregano from Cyprus area could significantly differ from those in other regions, which could influence antimicrobial activity.
Identifying the chemical compounds (or specific composition) with an anti-QS activity unique to the Cyprus area plants could be of interest, however, this was not studied in the present work.
Reviewer 3 Report
In this manuscript, the inhibition effects of ethanolic extracts of aromatic plants from Cyprus flora on microbial QS system were investigated. The research is interesting and the manuscript is well written. It can be accepted after revision. The specific comments are as follows.
(1) The major composition of the ethanolic extracts of aromatic plants, and the possible major components of the extracts that have anti-QS effects should be introduced. The possible mechanism of the anti-QS effects of the active components contained in the extracts should be introduced and discussed.
(2) It’s not proper for the sections of 2.2 and 2.4 to have the same sub-title name: “Effect of Plant Extracts on Biofilm Formation”. The contents of 2.4 can be moved to 2.2.
(3) Line 105, “cel-free” should be “cell-free”.
(4) The Latin strain names should be in italic type.
Reviewer 4 Report
Dear Author, I reviewed the manuscript (plants-1858476) entitled Anti-quorum Sensing Potential of Ethanolic Extracts of Aromatic Plants from Cyprus Flora. This manuscript presents relevant information about the anti-qs activity of Cyprus Flora plant's ethanolic extracts. However, some sections of the presented data can be improved. For this reason, I consider that this manuscript needs minor changes to be considered for publication in this journal.
Additional comments.
Highlight the advantages of using Cyprus Flora plants to inhibit the QS of pathogenic bacteria.
Check the paragraph extension in this manuscript.
Include an experimental design containing statistical factors and response variables in the statistical analyses applied to the findings of this research.
Compare the obtained findings with similar assays where natural extracts were tested to inhibit virulence factors and qs activity.
Include future trends to keep working with the obtained data.
Try to conclude with a general statement of the most relevant part of this study.
Author Response
Please see the attachement.

Round 2
Reviewer 2 Report
The authors now identified some active compounds, unfortunately without providing details, in particular - percentage composition.
The percentage composition of major components of the extracts should be provided. These data should be compared with each other and with the composition of similar extracts in other published works. For now, there is no evidence that the plants tested in this work significantly differ from the corresponding plants in other parts of the world. This hypothesis should be confirmed by analysis of the composition. However, the authors do not consider this part being within the scope of the present work...
Minor comment.
It's not clear how the bactericidal activity was determined, i did not find the procedure in Materials and Methods. According to the data in the Supplement, 1 or 2 mg/mL did not significantly reduce the growth of bacteria. However, according to the literature (e.g., PMID: 11041135 ), these concentrations for similar plants can already exhibit bactericidal activity. I suppose MIC and MBC should be determined for the examined extracts, and the differences (if there are some) with the published literature shoud be also discussed.
Reviewer 3 Report
The authors have revised accordingly. The revisions are acceptable, and it can be accepted after minor revisions.
"[19]" in line 111, and "[31]" in line 186 should not be italic.
Changes the first letter of "Rosmarinic" in figure 5 to lower case.